# Validation of the Decontamination of a Specialist Transport System for Patients with High Consequence Infectious Diseases

**DOI:** 10.3390/microorganisms9122575

**Published:** 2021-12-13

**Authors:** Claire Bailey, Catherine Makison-Booth, Jayne Farrant, Alan Beswick, John Chewins, Michael Eimstad, Fridtjof Heyerdahl, Brian Crook

**Affiliations:** 1Health and Safety Executive Science Research Centre, Buxton SK17 9JN, UK; claire.bailey@hse.gov.uk (C.B.); makisonbooth@who.int (C.M.-B.); jayne.farrant@hse.gov.uk (J.F.); alan.beswick@hse.gov.uk (A.B.); 2World Health Organization, 1211 Geneva, Switzerland; 3Health and Safety Executive, Leeds LS11 9AT, UK; 4Bioquell Ltd., Andover SP10 3TS, UK; john.chewins@bioquell.com; 5EpiGuard, 1634 Gamle Fredrikstad, Norway; michael.eimstad@epiguard.com (M.E.); fridtjof@epiguard.com (F.H.); 6Department of Prehospital Services, Oslo University Hospital, 0372 Oslo, Norway; 7Institute of Clinical Medicine, University of Oslo, 0315 Oslo, Norway

**Keywords:** infection, hospital, patient transport, decontamination, hydrogen peroxide, validation

## Abstract

When transferring highly infective patients to specialist hospitals, safe systems of work minimise the risk to healthcare staff. The EpiShuttle is a patient transport system that was developed to fit into an air ambulance. A validated decontamination procedure is required before the system can be adopted in the UK. Hydrogen peroxide (H_2_O_2_) vapour fumigation may offer better penetration of the inaccessible parts than the liquid disinfectant wiping that is currently suggested. To validate this, an EpiShuttle was fumigated in a sealed test chamber. Commercial bacterial spore indicators (BIs), alongside organic liquid suspensions and dried surface samples of MS2 bacteriophage (a safe virus surrogate), were placed in and around the EpiShuttle, for the purpose of evaluation. The complete kill of all of the BIs in the five test runs demonstrated the efficacy of the fumigation cycle. The log reduction of the MS2 that was dried on the coupons ranged from 2.66 to 4.50, but the log reduction of the MS2 that was in the organic liquids only ranged from 0.07 to 1.90, confirming the results of previous work. Fumigation with H_2_O_2_ alone may offer insufficient inactivation of viruses in liquid droplets, therefore a combination of fumigation and disinfectant surface wiping was proposed. Initial fumigation reducing contamination with minimal intervention allows disinfectant wipe cleaning to be completed more safely, with a second fumigation step inactivating the residual pathogens.

## 1. Introduction

In the UK, patients with high consequence infectious diseases (HCID), such as viral haemorrhagic fevers that are transmissible by contact and airborne transmissible monkeypox, may be admitted for initial treatment in an infectious diseases unit (IDU) at any hospital. Following their preliminary assessment, it may be considered necessary to transfer them to one of the hospitals in the HCID specialist network. Safe systems are therefore required in order to facilitate this transfer, with minimal risk to the healthcare staff while also considering patient comfort. An example arose in 2014–2015 when a healthcare worker (HCW) who was returning from working in an Ebola Treatment Centre in West Africa had taken ill and was initially treated at a hospital in Glasgow, before being transferred to the High Level Isolation Unit at Royal Free London [1]. More recently, a case of monkeypox in the UK, which was initially treated at a regional hospital, led to infections in the HCWs who were treating the index patient, emphasizing the need for good infection control and safe patient transfer [2]. The transfer of the Ebola patient necessitated the use of a portable Trexler flexible film isolator in order to maintain the patient’s care while protecting the healthcare staff from cross-infection. However, due to the size of the Trexler unit, this transfer involved using a military aircraft and a larger ambulance (known as a ‘jumbulance’) to move the unit from the airbase to the hospital.

EpiGuard^®^, a Norwegian company, has developed EpiShuttle^®^ as a safe means to transport patients with HCID. The EpiShuttle comprises a solid plastic base supporting a bed frame with a removable mattress mounted on plastic bed plates with a clip-on transparent plastic lid (known as the ‘hardtop’) (Figure 1). The unit has an integral P3 filtration unit in order to maintain ventilation and it can be operated at negative or positive pressure as appropriate. It has access ports with integral gauntlets for the safe administration of patient care and other ports for waste removal systems and the accommodation of IV lines or patient monitoring cables. While large enough to accommodate most patients (up to 198 cm in height and 150 kg in weight), the EpiShuttle is compact enough to fit in a standard-sized ambulance vehicle or a medium-sized air ambulance, and can be mounted on patient trolley undercarriage systems.

It has been planned for the EpiShuttle to be used for infected patient transfer in the UK National Health Service (NHS) via the Hazardous Area Response Teams (HART). Consequently, there needs to be a safe system of use implemented, including the control of any potential cross-infection from a patient to the HCWs during the placement of the patient into the EpiShuttle (for example, in a regional hospital), removal of the patient from the EpiShuttle at the receiving HCID Network specialist hospital, and subsequent decontamination of the EpiShuttle.

This paper describes a study that aims to validate the decontamination procedure that is applied to the EpiShuttle after each use. While EpiGuard have a manual decontamination procedure for the EpiShuttle that involves wiping with liquid disinfectant, a less ‘hands-on’ method would involve using a hydrogen peroxide (H_2_O_2_) vapour fumigation system [3,4], which may also ensure enhanced penetration to all of the parts of the equipment. HCID Units routinely use H_2_O_2_ vapour fumigation for other decontamination requirements [5], but they require validation for its use with the EpiShuttle.

This study was conducted shortly before the current COVID-19 pandemic. While the focus of the study was on high consequence infectious diseases leading to the potential release of large volumes of infected body fluids, an assessment is also included as to the potential use of the EpiShuttle specifically for COVID-19 patients’ transfer to hospitals from remote locations.

## 2. Materials and Methods

A 34 m^3^ stainless steel-lined controlled environment test chamber (with floor dimensions measuring 4.1 m × 2.55 m) at HSE’s laboratory was used to represent a sealable room at an HCID hospital. Once the air that was flowing through the room had achieved the required temperature (22 °C) and relative humidity (RH; 45% was the desired amount) settings in order to mimic the ambient conditions that are found in a hospital environment, the airflow was stopped and the inlet and outlet dampers were closed in order to seal the room so that the internal conditions were maintained. The fumigant was then introduced.

An H_2_O_2_ vapour fumigation system (Bioquell ProteQ, Andover, UK) was used, this system was capable of delivering the required concentration of the fumigant in order to achieve the minimum required criteria [6] for the successful virus (or phage, including that which was used as a surrogate) decontamination of a 4-log viral reduction (Figure 2).

An EpiShuttle unit was loaned to HSE by EpiGuard for testing purposes. A realistic post-use decontamination protocol was agreed with EpiGuard and Bioquell. The circular ports (EpiPorts) in the hardtop, as shown in Figure 1, house gauntlets that are each sealed when not in use between a removable lid on the outside and another inside. By removing the outside lid, a gauntlet can be accessed and, through it the inner lid, it can be removed so that the gauntlet can be used for patient care. It was assumed that at least two gauntlets would have been used during the patient transfer, therefore having these exposed for fumigation was deemed appropriate.

If a spill of contaminated body fluid occurred into the base of an EpiShuttle during patient transfer, it would be unrealistic to expect the fumigation to sufficiently penetrate into that fluid. Therefore, following the method that is described in the EpiShuttle Owner’s Manual, an initial wipe was conducted by staff wearing suitable personal protective equipment, using disposable absorbent cloths and a peracetic acid-based disinfectant (PeraSafe, Earls Colne, UK). During this procedure, all disposable items, such as the three mattress cushions, were safely transferred into clinical waste bags and the re-usable items, such as the three mattress support plates and the EpiPort lids, were placed into a plastic tray for subsequent soaking in disinfectant.

To maximise the fumigant’s penetration within the whole-room treatment configuration that is likely to be used in a hospital setting, it was agreed that the EpiShuttle base would be placed on the floor of the test chamber with the hardtop stood on its end, leaned against a wall (Figure 3 and Figure 4). In this way, the gauntlets would hang down, facilitating the exposure of their external surfaces, which are most likely to be contaminated, to the fumigant. Materials that were likely to absorb the fumigant, such as the cushions, were removed.

The challenge organisms that were used in this study were the MS2 bacteriophage and commercially pre-prepared *Geobacillus stearothermophilus* bacterial spore indicators (BIs) on stainless steel discs (Bioquell HPV-BI). Bacteriophages are easily propagated and assayed on a bacterial lawn, rather than requiring tissue culture. They are non-hazardous to humans and commonly employed as surrogates for other, more hazardous, viruses, such as those causing viral haemorrhagic fevers [7]. MS2 is one of the most popular surrogates for human-pathogenic viruses, it is used extensively in virus survival and virus inactivation studies [8].

Any cross-contamination in the EpiShuttle after patient use is likely to be in the form of splashes or droplets that have dried onto its surfaces, or in liquid residues that remain after wiping up a major spill of body fluid. Therefore, to provide a realistic challenge, the MS2 was either dried onto stainless steel coupons or suspended in small volumes of an aqueous solution (0.75 mL) with organic content (PBS buffer) designed to mimic blood, sputum or vomit. This has been achieved in previous work by the authors by adding 0.3% Bovine Serum Albumin (BSA) to the solution. The MS2 discs and suspensions were prepared to a titre of approximately 1 × 10^9^ plaque forming units (pfu) for Runs 1–4 in order to ensure clear discrimination beyond the required 4-log reduction. A 100-fold lower dilution of MS2 was used in Run 5.

The MS2 bacteriophages on steel coupons were placed in triplicate at the 6 representative locations in the EpiShuttle (as agreed with EpiGuard and Bioquell) and suspended in the aforementioned organic liquid, in triplicate, at a further 4 representative locations. The BIs, with a concentration of approximately 1 × 10^6^ colony forming units (cfu) per disc—such that is routinely used to test fumigation performance, were also placed in 3 locations in the EpiShuttle. The challenge test locations were laid out as summarised in Table 1 and shown in Figure 3 and Figure 4.

With the MS2 and BI biological challenges in place, the fumigation was undertaken with parameters that were calculated to match those that are typically used in hospital room and equipment fumigation, including the adjustment of the ambient temperature, humidity and dimensions of the test chamber. These parameters were 10 g/m^3^ of H_2_O_2_ vapour (340 g of H_2_O_2_ in total); a conditioning time of 5 min (300 s. as on system display); gassing at 16 g/min for 1250 s; a dwell time of 35 min (2100 s) for Runs 1–3; a 60 min dwell time (3600 s) for Runs 4 and 5; followed by an aeration time of 30 min (1800 s, using the test chamber room purge in combination with the aeration unit on the fumigation system). After completing each test run and venting the fumigant from the chamber, the biological challenges were retrieved for analysis.

After exposure to the fumigant, any surviving MS2 was removed from the discs by placing them into 10 mL of a nutrient broth and vortex mixing them for 1 min, then soaking them for 30 min. The resultant supernatant was serially diluted to 10^−5^ and 100 µL of each dilution was transferred into a separate plastic tube, to which 300 µL of a 3-h culture of the bacterium *E. coli* was added. This 400 µL of MS2/*E. coli* mixture was then added to 3 mL of semi-solid nutrient agar and immediately poured onto a pre-warmed nutrient agar plate in order to overlay the surface. These plates were left alone to allow the semi-solid agar to solidify before their overnight incubation, without inversion, at 35 °C. The growth of the *E coli* in the agar overlay created a cloudy appearance, with any resultant plaques (clear zones in the agar overlay) the result of MS2 bacteriophage growth lysing the bacteria. These were counted at a suitable dilution and the results were used in order to enumerate the surviving MS2, which was calculated as pfu per original carrier disc. The colony counts from the discs that were exposed to the fumigant were compared with the yield from the unexposed control discs in order to obtain a log reduction value. For the MS2 that was suspended in organic liquid, the starting material was serially diluted and then used to inoculate the plates (as described above) and then compared with the unexposed controls.

For the BI discs, the spores were washed into suspension using vortex mixing with sterile 2 mm-diameter beads and then serially diluted as described above; they were then used to inoculate the nutrient agar plates by the conventional spread plate method. These were incubated at 55 °C overnight and any resulting colonies were counted in order to calculate the cfu per BI disc, compared with the unexposed controls.

## 3. Results

### 3.1. Fumigation Parameters and Measured Hydrogen Peroxide Concentrations

Monitoring using the in-built sensors in the fumigation equipment confirmed that the pre-programmed parameters were met in order to deliver the concentration of H_2_O_2_ and dwell time that was calculated to meet the log kill requirements.

### 3.2. Overview of Fumigation Test Results

In total, five fumigation test runs were completed. Run 1 was used to scope the fumigation parameters, for which only BIs were used. Both the BIs and MS2 on discs and in liquid were exposed to the fumigant in Runs 2–5. Table 2 summarises the results and Figure 5 shows the log reduction data for the MS2 on discs and in liquid for Runs 2–5.

### 3.3. Log Reduction Results for BIs

In all five of the test runs, complete kill (>5.75 log reduction, compared with controls) was achieved for the *Geobacillus* spore BIs in each of the three locations in the EpiShuttle base (B1, near the foot end; B2, in the middle; and B3, near the head end).

### 3.4. Log Reduction Results for MS2 Bacteriophage Discs

No MS2 tests were conducted in Run 1. In Run 2, the average log reduction in the numbers of the MS2 pfu on the discs ranged from 3.34 to 3.88. The least log reduction on the discs occurred for the samples that were located at position 6, under the gauntlet. In Run 3, the average log reduction ranged from 3.64 to 4.50, the least log reduction being found in the samples that were located at position 2, in the base. In Run 4, with an extended dwell time but lower peak H_2_O_2_ concentration, the average log reduction ranged from 2.66 to 3.07, the least log reduction was found in the samples that were located at position 1, in the base. Run 5 was a repeat test run with the extended dwell time. Because incomplete kill was being achieved with the MS2 that was dried on the discs (as seen in Runs 2 to 4), it was speculated that this could be because the heavy inoculum had led to a lack of penetration of the fumigant and a ‘shielding’ effect of multiple layers of the bacteriophage, wherein the upper layers prevented the contact of the fumigant with the lower layers [9], further exacerbated by the presence of the organic content in the suspension medium that was used to prepare the discs. Therefore, for this run, the MS2 stock suspension that was used to prepare the discs was diluted to 1/100 of the concentration that was previously used. Despite this, the average log reductions of the MS2 on the discs still only ranged from 2.78 to 3.32, the least log reduction being found in the samples that were located at position 6, under the gauntlet. This indicated that there was a greater effect from the presence of the organic material than from the shielding effect described above.

### 3.5. Log Reduction Results for MS2 in Organic Liquid

No MS2 tests were conducted in Run 1. As shown in Figure 5, the log reductions were less in the liquid than on the discs. In Run 2, with the MS2 in the organic liquid, the average log reductions only ranged from 0.60 and 1.47. The least log reduction was found in the organic liquid samples that were located at position 9, in the base. In Run 3, the average log reductions were between 0.95 and 1.90, with the least log reduction found in the samples that were located at position 8, in the base. In Run 4, with an extended dwell time but lower peak H_2_O_2_ concentration, the average log reductions were between 0.07 and 0.37, with the least log reduction found in the samples that were located at position 10, at the foot of the hardtop. In Run 5, again with the extended dwell time and the liquid samples having been prepared from a 1/100 dilution of the MS2 stock suspension (as described above), the average log reductions were between 1.74 and 1.89, with least log reduction found in the samples that were located at position 9, in the base and under the mattress frame.

## 4. Discussion

H_2_O_2_ fumigation offers a highly practical and potentially effective means of decontaminating complex rooms, such as hospital wards or laboratories, as well as items of equipment [3,4], so long as these areas can be safely and effectively sealed off from their surroundings. Consequently, it is a viable option to decontaminate the multiple internal surfaces of EpiShuttle units after they have been used to transfer patients who are potentially suffering from HCID and, therefore, potentially shedding pathogens into their surroundings. It is a practical option in the UK as all of the specialist HCID units in hospital Trusts, and many other Trust hospitals, already use H_2_O_2_ decontamination systems and such facilities will be replicated across many countries. A whole-room approach is a pragmatic way to undertake this, provided a suitable, sealable room exists where the EpiShuttle components can be placed for treatment.

The test scenario that was used here was representative of healthcare facilities in which a sealable room can be used for H_2_O_2_ fumigation. Comparable parameters were used and the fumigation system was programmed to deliver the optimum concentration of the fumigant and dwell time for the size of room. The EpiShuttle hardtop was removed and stood on end so that the gauntlets hung loose in order to maximise their fumigant exposure and the absorbent materials were removed in order to prevent them from acting as absorption sinks for the fumigant.

This series of experiments used the MS2 bacteriophage as a test agent surrogate for infectious viruses and a standard bacterial spore BI that is routinely used to validate fumigation systems; the latter served as a useful indicator of the fumigant’s efficacy against a widely used spore. In all five test runs, the complete kill of the BI spores was achieved, demonstrating the efficacy of the fumigation cycle that was designed for the conditions. However, complete kill was not observed with any of the in-house prepared viral surrogate samples, demonstrating the importance of including relevant microbial challenges when carrying out validation of fumigation efficacy.

The MS2 bacteriophage challenge was presented both dried onto stainless steel coupons and in small volumes of liquid with high organic content. The former represents the potentially infective small-volume body fluid splashes that may dry onto the internal surfaces of the EpiShuttle. The latter aimed to represent the larger volumes of infective body fluid that may spill into the base of the EpiShuttle that cannot be wiped up as a part of the patient care that is carried out using the gauntlets during patient transfer. These challenges were considered appropriate by EpiGuard, the H_2_O_2_ fumigation system providers (Bioquell) and the end users, the UK’s NHS. The success criterion for fumigation was a 4-log reduction of MS2, this is a widely recognised measure for demonstrating disinfectant/fumigant efficacy and one that is stipulated within the European Test Standard [6].

It was anticipated that the liquid challenge would result in a less than 4-log reduction because previous work [10] has shown that fumigants struggle to penetrate liquid challenges that are used to mimic even relatively small spills. This proved to be the case, with the log reduction values in the liquid challenges consistently lower than 4 and ranging from 0.07 to 1.90. Previous work has also shown that fumigants may fail to completely penetrate drying liquid droplets that contain concentrated viral particles [11]. The fumigant’s performance against the MS2 that was dried on coupons was better, ranging from 2.66 to 4.50. In two of the four runs where an MS2 disc challenge was included, the lowest log reduction occurred in the samples that were located at position 6, under the gauntlet. It was anticipated that this could be a difficult location for the fumigant to penetrate, due to a shadowing effect, which was why care was taken to maximise the exposure by standing the hardtop on end so that gauntlets hung free. This, therefore, is an important practical finding to include in any recommended protocols.

There was no evidence that an extended fumigation dwell time (as tested in Runs 4 and 5) improved the log reduction. In Runs 2 and 3 with the MS2 on discs, the average log reduction across all of the locations ranged from 3.34 to 4.50, while in Runs 4 and 5 it ranged from 2.66 to 3.32. In Runs 2 and 3 with the MS2 in liquid, the average log reduction across all locations ranged from 0.60 to 1.90, while in Runs 4 and 5 it ranged from 0.07 to 1.89. Therefore, the shorter dwell time remains appropriate. There was some evidence of an improved log reduction with a lower bacteriophage titre in the test liquid (Run 5).

The results identify that the MS2 phage is inactivated when exposed to hydrogen peroxide, but within the experimental conditions the reduction was limited to, on average, 3.5 log. This suggests that the hydrogen peroxide vapour has been prevented from contacting a proportion of the phage, most likely due to the presence of the soiling substance.

## 5. Conclusions

For decontaminating complex equipment such as EpiShuttle patient transfer units, it is highly likely that H_2_O_2_ fumigation would be effective in killing human pathogens that cause high consequence infectious disease, based on the data obtained here by using a robust bacteriophage surrogate. The process is effective if these microorganisms are dried onto the surfaces and it has the advantage of penetrating into places that could be hard to reach by manual disinfectant wiping. While manual disinfection, in principle, could be equally effective, it can present practical challenges, including an additional risk of cross-infection via operator exposure, deviation from the manufacturer’s recommended concentrations and contact time instructions, and the potential for surfaces to be missed or insufficient product to be applied per unit area. However, H_2_O_2_ fumigation is less efficient when larger volumes of biological fluid are still present on surfaces, as a consequence of the limited penetration as shown in the extreme challenges presented in these experiments. A combination of H_2_O_2_ fumigation followed by disinfectant surface wiping may therefore be a viable option and also offer benefits in terms of protecting the staff from possible viral exposure, yet still achieve high levels of disinfection after the process’ completion.

H_2_O_2_ fumigation is used in many high containment microbiology laboratories after a major spill of infective material. UK laboratory biosafety guidance [12] sets a precedent for how to deal with such an eventuality. In that situation, it is recommended that the evacuated laboratory is first fumigated, then suitably protected staff enter to wipe up the spill and remove gross organic contamination, then a second round of fumigation is employed to remove any residual microbiological contamination. Based on this approach, for post-patient transfer decontamination of the EpiShuttle, from the evidence of this study we recommend that staff wearing appropriate PPE should:Set up a sealable room for the remote operation of an H_2_O_2_ fumigation unit.Transfer the EpiShuttle unit (with the hardtop put back in place after the patient is removed) into the fumigation room.Unclip the hardtop, separate it from the base and stand it on end against the wall so that any used gauntlets hang loose in order to maximise fumigant penetration.Remove the disposable items from the base (i.e., the mattress and pillow; as these could absorb fumigant) and place them into a clinical waste bag for disposal by incineration. No other intervention is required at this stage, thereby removing the need for the initial disinfectant wiping that is advised in the current manual disinfection-only method.Seal the room so that it is ready for fumigation.Use the shorter dwell time fumigation parameters that were described in this paper (as adapted to the dimensions of the room being used) to deliver the first fumigation cycle.After venting the fumigant from the room, check that the fumigation unit sensors show that the residual H_2_O_2_ residue levels are below statutory exposure limits (as designated in the UK by Health and Safety Executive EH40, https://www.hse.gov.uk/pubns/books/eh40.htm, accessed on 12 October 2021) and it is safe to re-enter.Again, wearing appropriate PPE, undertake the validated wipe-down and spray-disinfection of the EpiShuttle, including the dismantling of all removable components.Remove and dispose of the used PPE.Repeat steps 5 to 7.Perform the final cleaning wipe-down of the EpiShuttle and then reassemble it.

This work was commissioned and conducted before COVID-19 disease developed into a global pandemic. Although the causative agent, SARS-CoV-2, is classed as a Hazard Group 3 pathogen [13,14], in the UK it is not designated an HCID, therefore COVID-19 patients are unlikely to be transferred to an HCID hospital. However, UK HART teams may use the EpiShuttle for COVID-19 patient transfer, for example from remote or island locations to the receiving hospital, as has been done in Germany [15] and Switzerland [16]. While fumigation would also be effective in these circumstances, given that the EpiShuttle would not be contaminated with large volumes of highly infective body fluid, it is more likely that a validated liquid disinfectant wipe decontamination procedure will be used.

## Figures and Tables

**Figure 1 microorganisms-09-02575-f001:**
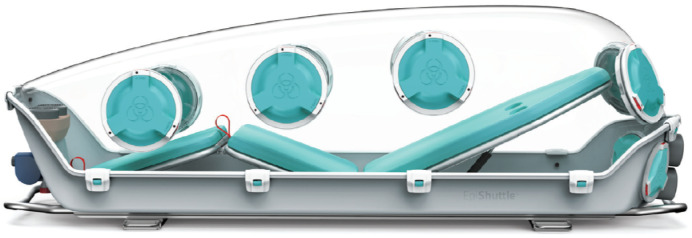
The EpiShuttle (image taken from EpiGuard’s EpiShuttle Owner’s Manual).

**Figure 2 microorganisms-09-02575-f002:**
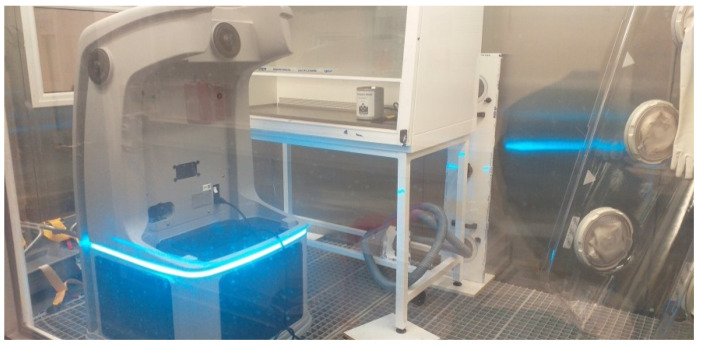
H_2_O_2_ fumigation system in test chamber.

**Figure 3 microorganisms-09-02575-f003:**
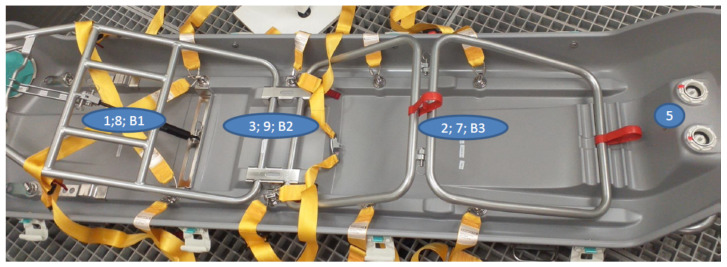
EpiShuttle base showing the numbered challenge test locations(1,2,3,5 = MS2 on coupons; 9 = MS2 in organic liquid; B1,B2,B3 = Biological Indicators).

**Figure 4 microorganisms-09-02575-f004:**
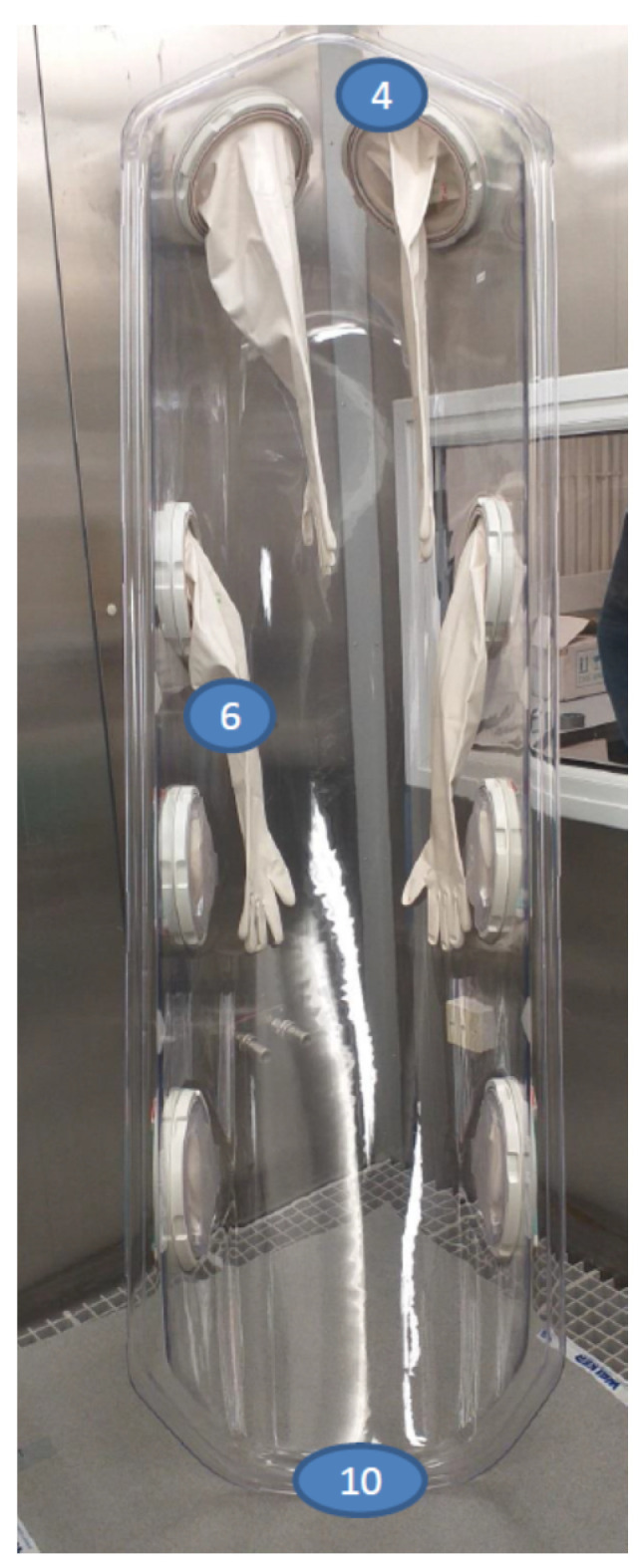
EpiShuttle hardtop showing the numbered challenge test locations (4,6 = MS2 on coupons; 10 = MS2 in organic liquid).

**Figure 5 microorganisms-09-02575-f005:**
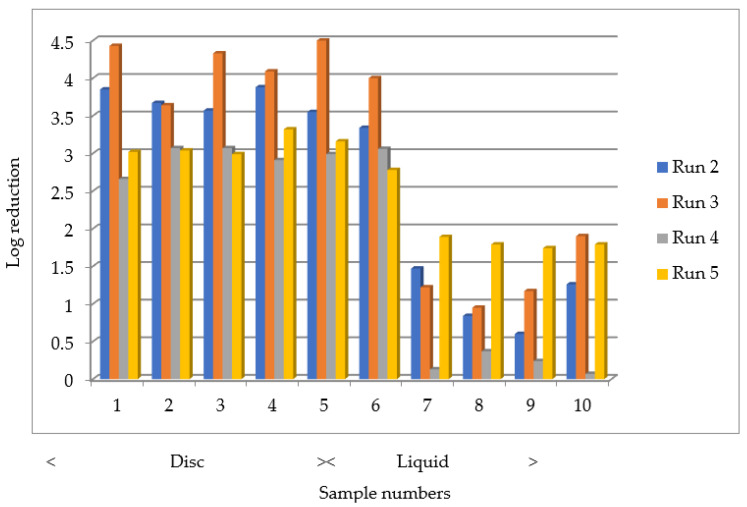
Log reduction of MS2 bacteriophage on discs and in organic liquid following H_2_O_2_ fumigation (data for BIs not included as all showed complete log reduction).

**Table 1 microorganisms-09-02575-t001:** Location of challenge test materials.

Location Description	Sample Number (Figure 3 and Figure 4)		
	MS2 on Steel Coupon	MS2 in Organic Liquid	*Geobacillus* Biological Indicator
EpiShuttle base near foot end	1	8	B1
EpiShuttle base in middle	3 (under frame)	9 *	B2 (under frame)
EpiShuttle base near head end	2	7	B3
EpiShuttle base adhered to inner wall at head	5		
Inside surface of EpiShuttle hardtop at head end	4		
Underside of extended gauntlet in EpiShuttle hardtop	6		
On floor at foot of EpiShuttle hardtop stood on end		10	

* Unable to fit under the frame so located close to frame.

**Table 2 microorganisms-09-02575-t002:** Reduction in numbers of MS2 bacteriophage and *Geobacillus* biological indicators.

Sample	Run 1		Run 2		Run 3		Run 4		Run 5	
	Av pfu	LogR	Av pfu	LogR	Av pfu	LogR	Av pfu	LogR	Av pfu	LogR
Discs										
1			1.16 × 10^4^	3.85	2.27 × 10^3^	4.43	4.52 × 10^6^	2.66	6.00 × 10^2^	3.02
2			1.75 × 10^4^	3.67	1.38 × 10^4^	3.64	1.75 × 10^6^	3.07	5.67 × 10^2^	3.04
3			2.20 × 10^4^	3.57	2.87 × 10^3^	4.33	1.74 × 10^6^	3.07	6.33 × 10^2^	2.99
4			1.08 × 10^4^	3.88	4.93 × 10^3^	4.09	2.55 × 10^6^	2.91	3.00 × 10^2^	3.32
5			2.33 × 10^4^	3.55	1.90 × 10^3^	4.50	2.13 × 10^6^	2.99	4.33 × 10^2^	3.16
6			3.75 × 10^4^	3.34	6.03 × 10^3^	4.00	1.79 × 10^6^	3.06	1.03 × 10^3^	2.78
Disc control			8.23 × 10^7^	N/A	6.07 × 10^7^	N/A	2.06 × 10^9^	N/A	6.23 × 10^5^	N/A
Liquids										
7			3.97 × 10^6^	1.47	1.66 × 10^7^	1.22	3.46 × 10^9^	0.13	1.27 × 10^4^	1.89
8			1.71 × 10^7^	0.84	3.11 × 10^7^	0.95	2.00 × 10^9^	0.37	1.60 × 10^4^	1.79
9			2.97 × 10^7^	0.60	1.87 × 10^7^	1.17	2.70 × 10^9^	0.24	1.79 × 10^4^	1.74
10			6.43 × 10^6^	1.26	3.50 × 10^6^	1.90	4.00 × 10^9^	0.07	1.60 × 10^4^	1.79
Liquid control			1.17 × 10^8^	N/A	2.78 × 10^8^	N/A	4.65 × 10^9^	N/A	9.80 × 10^5^	N/A
Biological indicators										
B1	ND	6.00	ND	6.07	ND	6.01	ND	5.75	ND	6.22
B2	ND	6.00	ND	6.07	ND	6.01	ND	5.75	ND	6.22
B3	ND	6.00	ND	6.07	ND	6.01	ND	5.75	ND	6.22
Control	9.92 × 10^5^	N/A	1.19 × 10^6^	N/A	1.02 × 10^6^	N/A	5.65 × 10^5^	N/A	1.65 × 10^6^	N/A

Av pfu = average plaque forming units (three replicates at each location); LogR = Log reduction; ND = none detected (lower limit of detection 50 cfu/disc) resulting in maximum log reduction based on control cfu/disc.

## Data Availability

The data presented in this study are available on request from the corresponding author. The data are not publicly available due to commercial confidentiality.

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
