# Peer review of "Validation of the Decontamination of a Specialist Transport System for Patients with High Consequence Infectious Diseases"

_microorganisms, 2021, doi:10.3390/microorganisms9122575_

Round 1

Reviewer 1 Report

A very well written article. Relevant content included, especially during a virus pandemic. The methodology of microorganism determination is presented correctly. It is not easy to perform such quantification of microorganisms from the surface. The methodologies described in the literature do not always allow to obtain reliable results. The results are presented correctly, the graphic form would be easier to perceive.

Author Response

I have submitted revised versions of the manuscript with and without the addition of a graphical presentation of the MS2 data (Figure 5 page 7) to see whether the Editors think the addition adds value to the paper.  If they do not think it does, there is a version of the manuscript without the Figure.

Reviewer 2 Report

This is a good paper.

I have just a few comments:

  1. Should EpiGuard have a ‘TM’ trademark symbol after it?
  2. ‘Covid’ should be written ‘COVID’
  3. The reference to triplicate locations on page 4 will be due to the rogue BI phenomenon. Please add a reference: Sandle, T. (2020) Rogue Biological Indicators: Are They A Real Phenomenon?, Journal of Validation Technology, 26 (1): DOI: https://www.ivtnetwork.com/article/rogue-biological-indicators-are-they-real-phenomenon
  4. On page 4, with the fumigation process, were any specific temperature or humidity requirements needed?

Author Response

Response from authors in italics:

  1. Should EpiGuard have a ‘TM’ trademark symbol after it?  Response - done – I checked the EpiShuttle owner’s manual and they have the ® symbol so I have added that to the first reference to EpiGuard and EpiShuttle on page 2.
  2. ‘Covid’ should be written ‘COVID’.  Response - This looks to have been corrected already by the Editorial team.
  3. The reference to triplicate locations on page 4 will be due to the rogue BI phenomenon. Please add a reference: Sandle, T. (2020) Rogue Biological Indicators: Are They A Real Phenomenon?, Journal of Validation Technology, 26 (1): DOI: https://www.ivtnetwork.com/article/rogue-biological-indicators-are-they-real-phenomenon Response - The paper by Sandle specifically deals with anomalies arising from incomplete kill seen with some outlier results from the use of commercial spore-based BIs, which is not relevant to the commercial BIs we used and the results we obtained with them.  However, it does refer to the complication of shielding effect caused by overlaying of biological material especially on in-house prepared test coupons, therefore we have included reference to this paper in section 3.4 discussing the MS2 bacteriophage disc results.  We thank the reviewer for bringing this paper to our attention.   
  4. On page 4, with the fumigation process, were any specific temperature or humidity requirements needed? Response - done - the fumigation equipment automatically accounts for such environmental parameters, so for completeness on page 4 we have added a reminder that this is taken into account (yellow highlighted text).